# Molecular Features of Metaplastic Breast Carcinoma: An Infrequent Subtype of Triple Negative Breast Carcinoma

**DOI:** 10.3390/cancers12071832

**Published:** 2020-07-08

**Authors:** Silvia González-Martínez, Belén Pérez-Mies, Irene Carretero-Barrio, María Luisa Palacios-Berraquero, José Perez-García, Javier Cortés, José Palacios

**Affiliations:** 1Clinical Researcher, Hospital Universitario Ramón y Cajal, 28034 Madrid, Spain; silviagonzalezmartinezbio@gmail.com; 2Pathology Department, Hospital Universitario Ramón y Cajal, 28034 Madrid, Spain; bperezm@salud.madrid.org (B.P.-M.); irene.carretero@salud.madrid.org (I.C.-B.); 3Instituto Ramón y Cajal for Health Research (IRYCIS), 28034 Madrid, Spain; 4CIBER-ONC, Instituto de Salud Carlos III, 28029 Madrid, Spain; 5Faculty of Medicine, University of Alcalá de Henares, Alcalá de Henares, 28801 Madrid, Spain; 6Breast Pathology Unit, Hospital Universitario Ramón y Cajal, 28801 Madrid, Spain; 7Hematology and Hemotherapy Department, Clínica Universidad de Navarra, 31008 Pamplona, Spain; mpalaciosb@unav.es; 8IOB Institute of Oncology, Quironsalud Group, Hospital Quiron, 08023 Barcelona, Spain; jose.perez@medsir.org; 9IOB Institute of Oncology, Quironsalud Group, 28006 Madrid, Spain; 10Medica Scientia Innovation Research, 08018 Barcelona, Spain; 11Medica Scientia Innovation Research, Ridgewood, NJ 07450, USA; 12Vall d’Hebron Institute of Oncology, 08035 Barcelona, Spain

**Keywords:** MBC, metaplastic breast carcinoma, molecular alterations, prognosis, treatment

## Abstract

Metaplastic breast carcinoma (MBC) is a heterogeneous group of infrequent invasive carcinomas that display differentiation of the neoplastic epithelium towards squamous cells and/or mesenchymal-type elements. Most MBC have a triple negative phenotype and poor prognosis. Thus, MBC have worse survival rates than other invasive breast carcinomas, including other triple negative breast carcinomas (TNBC). In this study, we reviewed the molecular features of MBC, pointing out the differences among subtypes. The most frequently mutated genes in MBC were *TP53* and *PIK3CA*. Additionally, mutations in the other genes of the PI3K/AKT pathway indicated its importance in the pathogenesis of MBC. Regarding copy number variations (CNVs), *MYC* was the most frequently amplified gene, and the most frequent gene loss affected the *CDKN2A/CDKN2B* locus. Furthermore, the pattern of mutations and CNVs of MBC differed from those reported in other TNBC. However, the molecular profile of MBC was not homogeneous among histological subtypes, being the alterations in the PI3K pathway most frequent in spindle cell carcinomas. Transcriptomic studies have demonstrated an epithelial to mesenchymal program activation and the enrichment of stemness genes in most MBC. In addition, current studies are attempting to define the immune microenvironment of these tumors. In conclusion, due to specific molecular features, MBC have a different clinical behavior from other types of TNBC, being more resistant to standard chemotherapy. For this reason, new therapeutic approaches based on tumor molecular characteristics are needed to treat MBC.

## 1. Introduction

Among predominantly triple negative breast carcinomas (TNBC), metaplastic breast carcinoma (MBC) is a histologically heterogeneous group of invasive carcinomas that display heterologous differentiation of the neoplastic epithelium towards squamous cells and/or mesenchymal-type elements such as spindle, chondroid, and osseous cells [1]. Although it is a rare subtype of breast cancer, accounting only for 0.2–5% of invasive breast carcinomas [1], MBC is of considerable interest not only because of its pathological heterogeneity, but also because of its differences in clinical behavior compared to conventional invasive carcinoma and other types of TNBC [2]. In general, MBC is more resistant than other TNBC to conventional chemotherapy. For this reason, new therapeutic approaches are needed. In this article, we reviewed the main molecular features of this group of tumors, pointing out the differences among MBC subtypes. 

## 2. MBC Subtypes

According to their behavior and histopathological features, MBC can be subclassified as high-grade and low-grade [3]. High-grade MBC (HG-MBC) include squamous cell carcinoma (SqCC), spindle cell carcinomas (SpCC), and MBC with heterologous mesenchymal differentiation (MBCHMD). In addition, mixed MBC (MMBC) is composed of more than one subtype (Figure 1). Low-grade MBC are represented by low-grade adenosquamous carcinoma (LGASC) and fibromatosis-like MBC (FLMBC) [1]. In this review, we have focused on HG-MBC. 

SpCC is characterized by a predominant proliferation of intermediate to highly atypical spindle cells adopting multiple architectural patterns. Squamous differentiation can be predominant in some MBC or may be present in different proportions with other components, especially spindle cells, in MMBC. Pure SqCC are frequently cystic tumors, in which a central cavity lined by atypical squamous cells is surrounded by neoplastic cells, with different degrees of squamous differentiation and a reactive stroma [1]. MBCHMD is composed of an admixture of mesenchymal components, more frequently cartilage and bone, with carcinomatous areas that can present glandular or squamous differentiation [1,4]. This subtype has also been denominated as matrix-producing MBC. 

## 3. Clinicopathological Features

MBC is more frequently diagnosed in women between 50 and 60 years old, with a median age around 59 years. In one series, important age differences among histological subtypes were observed, MBC with chondroid differentiation occurring at a mean age of 71 years, whereas spindle and squamous cell carcinomas presented at mean ages of 56 and 48 years old, respectively [5]. MBC are usually large tumors with a mean size of 3.9 cm, among the series reviewed [5,6,7,8,9,10,11,12,13,14,15,16,17,18,19,20,21,22,23,24,25,26,27]. Most tumors present in stage II. About 35% of the tumors have lymph node metastases and 13% visceral metastasis at diagnosis. Lymphovascular invasion is observed in nearly 20% of the tumors. 

Regarding biomarker expression, around 85% of HG-MBC are triple negative. Hormone receptor positivity and *HER2* positivity has been reported in 0–13% and 0–10% of the series, respectively [5,6,7,8,9,10,11,12,13,14,15,16,17,18,19,20,21,22,23,24,25,26,27].

## 4. Molecular Alterations

In this study, we reviewed 14 series [5,6,7,8,9,10,11,12,13,14,15,16,28,29] including a total of 539 molecularly characterized tumors. Eight series [5,6,9,10,11,15,29,30] included copy number variations (CNVs) data. The most frequent alterations are presented in Appendix A, which includes the specific results of all the studies.

*TP53* was the most frequently mutated gene in the 13 series that sequenced it, and it was found altered in all histological subtypes [5,6,7,8,9,10,11,12,13,14,15,29,31]. The frequency of mutations ranged from 26% to 70%, (median 58.7%). Around 100 different mutations were observed in the 224 cases of 8 series in which the mutations were detailed [5,7,8,10,11,12,13,15]. Among all the observed mutations, the most frequent affected R273. In 8 tumors, the R273H variant was observed (3.6%) and in 6 tumors, the R273C (2.7%) was observed (Appendix A). R273 is a hotspot mutation with the R273H, R273C, and R273G variants occurring most commonly on patient samples. R273H and R273C lead to a more aggressive phenotype than R273G [32].

*PIK3CA* was the second gene with the highest mutation frequency. This gene was sequenced in 13 of the 14 series reviewed [5,6,7,8,9,10,11,12,13,14,15,29,31], and the frequency of mutations ranged from 12% to 48%, (median 32.8%). Nineteen different mutations were observed in the 224 cases of the 8 series in which the mutations were detailed [5,7,8,10,11,12,13,15], being 17 of the mutations missense substitutions and only 2 frameshift mutations. The most characteristic mutation was H1047R, which occurred in 43 cases (19%) (Appendix A). H1047R is a common hotspot driver mutation in human breast cancers that causes a constitutive activation of the PI3K protein and is associated with disease progression and poor prognosis [33,34]. The next most frequently observed mutation was H1047L in 7 cases (3%) (Appendix A). 

Mutations in other genes of the PI3K/AKT pathway indicated the importance of this pathway in the pathogenesis of MBC [31]. In the revised series, in addition to mutations in *PIK3CA*, mutations in *PTEN* (12.7%), *PIK3R1* (11.2%), *NF1* (9.8%), *HRAS* (8.5%), and *AKT1* (3%) also occurred (Appendix A). Interestingly, simultaneous mutations in more than one gene of the pathway can occur, such as the simultaneous mutations in *HRAS* and *PIK3CA*, which have also been recognized as driver mutations in both benign and malignant breast adenomyoepitheliomas [3,35].

Somatic mutations in *NF1*, a gene involved in the PI3K/AKT pathway, have been detected in nearly 10% of MBC (Appendix A). Interestingly, there have been reports of the development of MBC in 6 patients with type 1 neurofibromatosis (*NF1*) [36,37,38,39,40,41]. Patients ranged in age from 41 to 57 years and the tumors presented in stages II or III, measuring between 3 and 10 cm. Squamous differentiation was present in half of the patients, and the rest of the cases showed a conspicuous spindle cell component. 

Krings et al. [11] demonstrated for the first time that MBC frequently carries critical point *TERT* promoter mutations, which are responsible for upregulation of *TERT* expression and telomerase activation. These *TERT* promoter mutations occur in 25–33% of MBC [9,11] (Appendix A).

Although a study reported a high frequency of *CTNNB1* (beta-catenin) mutations in MBC [16], subsequent series have not confirmed this finding. However, it is thought that the WNT pathway can play a role in the development of some MBC [12,16,31]. In this sense, some series have demonstrated gene mutations that modulate the WNT pathway, such as *APC* (5%), a key component of the beta-catenin degradation complex, and *FAT1* (11%) (Appendix A). Moreover, it is well established that *TERT* promoter mutations can activate the WNT pathway, among others, through telomere-independent mechanisms [42].

It has been shown that some genes involved in DNA repair, including *BRCA1,* are downregulated in MBC compared to other TNBC [43]. MBC can carry mutations in *BRCA1* (3–15%), *BRCA2* (2–6%), and *ATM* (2–12%) (Appendix A). Interestingly, MBC have been occasionally reported in *BRCA1* germline mutation carriers [44,45,46,47].

Finally, MBC can also carry mutations in chromatin remodeling genes, such as *KMT2D* (17%) and *ARID1A* (6%), as well as in other genes at lower frequencies (Appendix A).

Regarding CNVs in MBC, *MYC* was the most frequently amplified gene, occurring in 17.3% of the tumors (Appendix A). *MYC* modulates several cellular functions, including proliferation. Other genes implicated in cell cycle control that were found amplified in more than one series of MBC included *CCND1* (8.4%), *CCNE1* (5.9%), and *CDK4* (4%). In addition, *CCND3* and *CCND2* were found amplified in 15% and 5% of MBC in a single study, respectively (Appendix A). 

The second gene with the highest frequency of amplification in MBC was *EGFR* (17.2%). Other amplified genes also coding for tyrosine kinase receptors included *FGFR1* (5%) and *ERBB2* (4.8%). Downstream pathways, such as *KRAS* and *PI3K*, were also affected by gene amplification in *RAS*, *NF1, PIK3CA, SOX2*, and *AKT3* (5% frequency in each gene) (Appendix A).

The most frequent gene loss affected the *CDKN2A/CDKN2B* locus (19%), *PTEN* (14.9%) and *RB1* (6.5%) losses were found at lesser frequencies (Appendix A).

The pattern of mutations and CNVs of MBC differed from those reported in other breast carcinoma types. Thus, Table 1 and Table 2 show a comparison of the main molecular alterations of MBC compared with invasive ductal carcinoma (IDC) with a basal like gene expression profile in the TCGA series, and TNBC in the MSKCC series [29].

All these data tend to suggest that *TP53* is less frequently mutated in MBC that in other TNBC, but genes of the *PIK3CA* are more frequently mutated in this group of TNBC. However, these differences cannot apply to all histological MBC subtypes because differences in their molecular profile exist. Thus, Table 3 shows the distribution of *TP53* and *PIK3CA* mutations in pure specific subtypes of MBC. *TP53* mutations were more prevalent in SqCC and MBCHMD than in SpCC. Mutations in *PIK3CA* predominated in SpCC, and SqCC was the only subtype that carried *PTEN* mutations (28%), also involved in the *PI3K* pathway [12,15]. In addition, Kring et al. [11] showed promoter *TERT* mutations in 80% of SpCC and in 20% of SqCC, but in none of the 10 MBCHMD. Amplifications of *MYC* and *EGFR* were significantly more frequently observed in tumors with squamous or spindle cell metaplasia [30]. 

## 5. Epithelial to Mesenchymal Transition in MBC

Epithelial to mesenchymal transition (EMT) is a biological process that involves the acquisition of a mesenchymal phenotype by the (malignant) epithelial cells, endowing these cells with migratory and invasive properties, promoting cancer progression, preventing cell death and senescence, and inducing resistance to chemotherapy [48,49,50]. Former studies by Lien et al. [51] and Sarrió et al. [48] showed that MBC displays a pattern of expression consistent with EMT. Subsequent studies have confirmed these observations. 

The acquisition of a mesenchymal phenotype in MBC is associated with a switching of cadherins (from E-cadherin to N-cadherin and cadherin 11) and the downregulation of epithelial markers including not only E-cadherin (*CDH1*), but also occludens zone 1 (*ZO1*), desmoplakin, keratin 7, keratin 18, and keratin 19, among others [43]. In contrast, genes related to the mesenchymal phenotype, motility, migration, and formation of extracellular matrix, such as vimentin, SPARC, or different collagen types, are upregulated in MBC [31,43,50,51,52].

EMT can be activated by different pathways, such as Transforming Growth Factor Beta 1 (TGFβ), tyrosine kinase receptors, Wnt, and Hippo, among others [53]. These pathways converge in the activation of EMT transcription factors (EMT-TF), such as *SNAI1* (Snail), *SNAI2* (Slug), *ZEB1*, *ZEB2, TWIST*, etc. Most of these EMT-TF, which function as transcriptional repressors of E-Cadherin, are upregulated in MBC [31,50,54,55,56].

The miR-200 family plays a major role in regulating epithelial plasticity, mainly through its involvement in double-negative feedback loops with the EMT-TF *ZEB1, ZEB2, SNAI1*, and *SNAI2*, ultimately influencing E-cadherin expression levels. The downregulation of the miR-200 family has been reported in MBC and is not only due to the transcriptional repression by EMT-TF, but also to the promoter methylation [55]. 

EMT in MBC allows cells to acquire different phenotypes due to secondary trans-differentiation. Accordingly, different expression patterns have been reported in SpCC, SqCC, and MBCHMD with chondroid differentiation [5,52]. Thus, SpCC showed the downregulation of epithelial genes, such as *CDH1, EPCAM, CLDN3,* and *CLDN8*. Accordingly, the transcriptomic profile of SpCC is closely related to that reported in claudin-low breast cancer [5]. In contrast, tumors with chondroid differentiation showed an upregulation of genes involved in chondrocyte differentiation and chondroid matrix maintenance [5]. Moreover, *OSMR*, which plays a role in mesenchymal differentiation and bone formation, was found to be mutated in two MBC with extensive osseous differentiation and in no other histopathological subtype [57].

Finally, MBC have markedly elevated CD44/CD24 and CD29/CD24 ratios and ALDH-1 expression, suggesting that MBC could be enriched in stem-cell-like cells [31,54,58].

## 6. Immune Microenvironment

There are few studies analyzing the immune microenvironment in MBC. Results on programmed cell death ligand-1 (PD-L1) expression in MBC tumor cells are conflicting, probably due to the low number of tumors analyzed to date. Joneja et al. [13] and Dill et al. [59] observed that 45% (39 out of 72) and 40% (2 out 5) of MBC in their respective series were positive. They defined PD-L1 positivity as expression in ≥5% and ≥1% of tumors cells, respectively. Vranic et al. [6] found that 33.3% of all SpCC expressed PD-L1 in cancer cells above the 1% threshold, 3 of them exhibiting diffuse PD-L1 expression in cancer cells (50–100% cancer cell positive). In contrast to these previous results, Zhai et al. [8] did not identify PD-L1 immunoreactivity in 18 MBC using the same antibody clone and assay platform that was used in the three previous studies.

At present, mechanisms responsible for PD-L1 expression remain unknown, since only one study has reported PD-L1 locus amplification in 0.5% of MBC (1 out of 192) [9]. However, it has been demonstrated that EMT contributes to evasion of immune surveillance in breast cancer [60] and that PD-L1 is upregulated in EMT-activated human breast cancer cells by a mechanism involving *ZEB-1* and miR-200 [61]. These observations might explain the upregulation of PD-L1 in MBC.

There is scarce information regarding inflammatory cells in MBC. Tray et al. [9], in a study of 20 tumors, reported that the median percentage of tumor-infiltrating lymphocytes (TILs) was 40 in 9 tumors with high tumor mutational burden (TMB), compared with 20 in tumors with low TMB (without statistically significant differences). It is important to note that only 9 out of the 192 MBC in this study showed high TMB.

In a study by Joneja et al. [13], 40% of the MBC (30 out of 70) showed high PD-1 expression in the inflammatory infiltrate. The authors considered high expression to occur when the number of PD-1-positive inflammatory cells was higher than the median (22.5; range 0–400). In this study, 23% of MBC showed both PDL-1 tumor expression and high PD-1 inflammatory cell expression.

Mismatch repair deficiency, a molecular characteristic associated with inflammatory infiltration in other tumors, is very infrequent or absent in MBC [9].

## 7. MBC Prognosis

MBC is a poor prognosis invasive breast cancer. According to our review, its prognosis was worse not only when compared with conventional invasive ductal carcinomas, but also when compared with other TNBC. We reviewed 8 series comparing survival between patients with MBC and TNBC [21,22,24,25,26,62,63,64]. These series included a total of 2,007 MBC and 41,937 TNBC. The five-year survival rate ranged from 44% to 65.3% in MBC and 78% to 90.6% in TNBC. Furthermore, survival in one of the series at 3 years was 74.2% in MBC and 80% in TNBC. Statistically significant differences were observed in all but one series, reporting only 24 cases of MBC [64].

In addition, we reviewed 12 series in which the survival rates between patients with MBC and other types of breast cancer, not specifically TNBC, were compared [20,22,23,26,27,28,64,65,66,67,68,69]. These series included a total of 8,154 MBC and 229,075 cases of non-MBC. The five-year survival rate ranged from 54.5% to 88.9% in MBC and 85.1% to 98.4% in other types of breast cancer. Survival in one of the series at 3 years was 76.7% in MBC and 92.4% in non-MBC. Statistically significant differences were observed in 10 of the series. The two series in which statistical differences were not observed included a low number of cases [28,64]. 

There are conflicting reports regarding whether the prognosis of HG-MBC differs among different histological groups. Rakha et al. [70] and McCart Reed et al. [7] observed in two large international multicenter series of MBC (*n* = 405 and *n* = 347) that SpCC and mixed histologies had more aggressive biological behaviors than SqCC and MBCHMD. However, other authors analyzing series with a smaller number of cases have not found statistically significant differences among these histological subgroups [19,20,21,22,24].

There are also conflicting reports regarding whether the prognosis of MBC differs depending on the receptor status. In a study by Mills et al. [65] analyzing 3,685 MBC of which 74.5% were TN, no differences in prognosis were found between TN and non-TN MBC. In contrast, in a study by Ong et al. [66] analyzing 2,451 MBC of which 70% were TN, a worse prognosis was observed in TN MBC when compared with ER-positive MBC, but no differences were observed when compared with HER2-positive MBC, which represented 4.8% of the cases.

Additionally, the series by Mills et al. [65] and Song et al. [68], also compared TN MBC with TN non-MBC. Song et al. found a statistically significant difference in the 5-year overall survival rate, being 54.5% in MBC and 73.3% in TN invasive ductal carcinoma. Mills et al. found that TN MBC was associated with inferior overall survival compared to other TN histologic subtypes, such as invasive ductal carcinoma.

It has been suggested that the poor prognosis in MBC might be due to a higher degree of chemoresistance than in other TNBC. In this sense, there are few studies analyzing pathological response to neoadjuvant chemotherapy in MBC. In 11 studies including a total of 150 patients [19,22,24,31,64,71,72,73,74,75,76], most of them using regimes based on anthracyclines and taxanes, a complete pathological response was observed in 11.3% (range 0–25%) of the patients, a frequency which is lower than the 34% reported in TNBC by Cortazar in a metanalysis of 12 clinical trials [77].

Although all studies included few cases and the sample is not powered to analyze differences among histological subgroups, these results clearly demonstrate that MBC shows poor chemosensitivity to conventional chemotherapy. 

## 8. Targeted Therapy in MBC

Conventional therapeutic approaches, including surgery, chemotherapy, and radiotherapy have been the focus of a recent review [78].

Due to its relative chemoresistance and poor prognosis, new approaches based on tumor molecular characteristics are needed to treat MBC. The high frequency of alterations in the PI3K/AKT/mTOR pathway makes MBC a good candidate to be treated with mTOR inhibitors. Accordingly, it has been reported that patients with advanced MBC treated with an mTOR-based systemic therapy regimen (temsirolimus or everolimus with liposomal doxorubicin and bevacizumab) had better long-term outcomes compared with patients with non-metaplastic triple-negative breast cancer treated with the same regimen, suggesting that metaplastic histology may predict benefits from agents targeting the PI3K/AKT/mTOR pathway. In addition, the authors observed that the presence of a PI3K pathway aberration was associated with a significant improvement in objective response rate [79,80].

Anecdotic observations have demonstrated that some MBC can respond to check-point inhibitors. Thus, Adams et al. [81] and Al Sayed et al. [82] reported complete responses to pembrolizumab and durvalumab, respectively, in two patients with advanced MBC with PD-L1 tumor cell expression. The DART study (NCT02834013) evaluates dual anti-CTLA-4 (ipilimumab) and anti-PD-1 (nivolumab) blockade in rare tumors, including an MBC cohort (Arm 36) [11]. Additionally, the FDA has approved therapy with atezolizumab for TNBC containing 1% PD-L1 positive immune cells in the tumor biopsy, based on the IMpassion130 clinical trial (NCT02425891) [8].

Two randomized phase III trials, the OlympiAD [83] and the EMBRACA [84], have reported PARP inhibitors’ (PARPi) efficacy in comparison with the physicians’ choice of chemotherapy for patients with locally advanced/metastatic breast cancer and a germline *BRCA* pathogenic variant (g*BRCA*-positive). As previously reviewed, some MBC can occur in g*BRCA*-positive women and can respond to PARPi. In this sense, Litton et al. [85] reported a complete pathological response of an operable breast cancer treated with talazoparib alone for 6 months in a patient with an MBC and a known germline *BRCA2* pathogenic variant. It would be interesting to determine if MBC with somatic mutations in *BRCA1* or *BRCA2* and with *BRCA1* promoter hypermethylation also respond to PARPi.

## 9. Conclusions

Our understanding of the molecular alterations of breast cancer have enormously advanced during the last two decades and have been the basis for the development of new targeted therapies. More recently, studies based on next generation sequencing are also deciphering the molecular profile of infrequent breast cancer histological types, such as MBC. Considering the poor response of MBC to conventional chemotherapy, new targeted therapies are urgently needed. Although some preliminary studies have shown promising results using m-TOR inhibitors and, in anecdotic cases, response to PARPi and check-point inhibitors, at present there are no targeted therapies specifically indicated for MBC. Although it is recommended that MBC should be included in clinical trials designed for TNBC, the differences in the molecular profiles between MBC and other TNBC, as well as the heterogeneity among MBC subtypes, will probably translate into a more complicated interpretation of results. Ideally, specific cohorts of different MBC subtypes should be included and specifically analyzed according to their molecular features in clinical trials. Due to the low prevalence of these tumors, multi-institutional efforts should be made in order to accelerate the identification of effective therapies for this aggressive tumor.

## Figures and Tables

**Figure 1 cancers-12-01832-f001:**
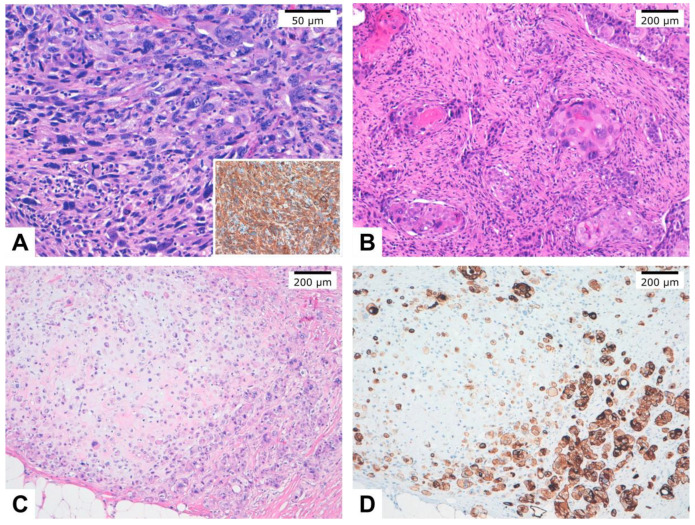
Histopathological images of different metaplastic breast carcinoma subtypes. (**A**). Spindle cell carcinoma entirely composed of neoplastic spindle cells. Inset: Immunohistochemistry showed cytokeratin AE1/AE3 diffuse and intense staining in tumor cells. (**B**). Squamous cell carcinoma with keratin pearls. (**C**). Metaplastic carcinoma with mesenchymal (chondroid) differentiation (matrix-producing carcinoma). (**D**). Immunohistochemistry of case C showed cytokeratin AE1/AE3 expression staining in tumor cells.

**Table 1 cancers-12-01832-t001:** Comparison of mutations of metaplastic breast carcinoma (MBC) compared with invasive ductal carcinoma (IDC) series.

Gene	MBC (series *)	TCGA (*n*= 153)	MSKCC (*n* = 46)
*TP53*	58.7% (13)	88.9%	87%
*PIK3CA*	32.8% (13)	6.5%	13%
*TERT*	29% (2)	-	-
*KMT2D*	17% (3)	2.6%	-
*PIK3R1*	11.2% (6)	3.3%	6.5%
*PTEN*	12.7% (10)	3.9%	4.3%
*RB1*	9.5% (4)	5.2%	8.7%
*NF1*	9.8% (5)	3.9%	6.5%
*HRAS*	8.5% (9)	0.7%	-
*ARID1A*	6% (3)	2%	4.3%
*GNAS*	4.6% (3)	-	-
*APC*	4.7% (4)	3.3%	-

* Total number of series in which the percentage of mutation in that gene is specified. Regarding CNVs, Table 2 shows the comparisons among series.

**Table 2 cancers-12-01832-t002:** Comparison of copy number variations (CNVs) of MBC compared with IDC series.

Gene	MBC (series *)	TCGA (*n* = 153)	MSKCC (*n* = 46)
*MYC* amplification	17.3% (4)	35.9%	17.4%
*CCNE1* amplification	5.9% (4)	12.4%	13%
*CCND1* amplification	8.4% (5)	2%	4.3%
*CDKN2A* deletion	19% (5)	7.8%	4.3%
*PTEN* deletion	14.9% (4)	17.6%	6.5%
*RB1* deletion	6.5% (3)	10.5%	2.2%

* Total number of series in which the percentage of amplification or deletion in that gene is specified.

**Table 3 cancers-12-01832-t003:** Distribution of *TP53, PIK3CA*, and *TERT* mutations in three different histological subtypes (from references [8,11,12,15]).

Histological Subtype	*TP53* *n* (%)	*PIK3CA* *n* (%)	*TERT* *n* (%)
Spindle cell carcinoma	8/26 (30.7)	10/21 (47.6)	4/5 (80)
Squamous cell carcinoma	23/27 (85)	9/27 (33.3)	1/5 (20)
Heterologous metaplastic carcinoma	19/27 (70)	4/27 (15)	0/10

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
