# Peer review of "Molecular Features of Metaplastic Breast Carcinoma: An Infrequent Subtype of Triple Negative Breast Carcinoma"

_cancers, 2020, doi:10.3390/cancers12071832_

Round 1

Reviewer 1 Report

The review article on metaplastic breast carcinoma is comprehensive with 85 references. 

Seven aspects of metaplastic breast  carcinoma were reviewed.

The article has a potential to be a reference article for future research on metaplastic breast cancer and triple negative breast cancer.

Author Response

The review article on metaplastic breast carcinoma is comprehensive with 85 references. 

Seven aspects of metaplastic breast carcinoma were reviewed.

The article has a potential to be a reference article for future research on metaplastic breast cancer and triple negative breast cancer.

We would like to thank the reviewer for your kind comments on our work.

Reviewer 2 Report

The authors reviewed characteristics of metastatic breast carcinoma (MBC) from various point of views including molecular alterations, EMT/stemness, Immune response and therapeutics. Thoug this review is well-organized and it will be very informative to the researchers in this field, the manuscript can be improve with a few revision. Here is my comments.

(1) Figure-1 needs a title of the figure.

(2) Figure 1C/D should be presented as Figure 1A

(3) Line 66: Please describe full name of HG-MBC

(4) Line 166: Please check the sentence

(5) Line 174-207:  EMT and stemness is different story. EMT is more related to metastasis and cancer stem cell is more associated with tumour growth and relapse after therapy. I suggest to review stemness or cancer stem cell part under another subtitle.

(6) Line 194: The well-known members of miR-200 family that are associated with EMT are miR-200a/b/c (Gregory PA et al. Nature Cell Biology. 2008). I am not sure why authors focus on miR-200f

(7) Line 224: Please check the sentence

(8) Line 237-241: According to line 50-51, MBC seems to be a subtype of TNBC. Then, is it appropriate to compare the patients survival between TNBC and MBC?

Author Response

We would like to thank the reviewer for the helpful suggestions and comments.

  • Figure-1 needs a title of the figure.

It has been added.

  • Figure 1C/D should be presented as Figure 1A

Figure 1 has been edited to be presented as such.

  • Line 66: Please describe full name of HG-MBC

The full name has been described on line 62.

  • Line 166: Please check the sentence

The sentence has been checked on line 187. As table 3 shows, TP53 mutations were more prevalent in squamous cell carcinoma (85%) and heterologous metaplastic carcinoma (70%) than in spindle cell carcinoma (30.7%).

  • Line 174-207:  EMT and stemness is different story. EMT is more related to metastasis and cancer stem cell is more associated with tumour growth and relapse after therapy. I suggest to review stemness or cancer stem cell part under another subtitle.

The EMT section has been edited accordingly (lines 195, 196, 197, 207, 208, 218). We have added a paragraph on stemness in line 227-228.

  • Line 194: The well-known members of miR-200 family that are associated with EMT are miR-200a/b/c (Gregory PA et al. Nature Cell Biology. 2008). I am not sure why authors focus on miR-200f

As was stated in line 213, miR-200f is the abbreviated name for miR-200 family. To avoid future misunderstandings, this abbreviation will not be used in the manuscript (line 215).

  • Line 224: Please check the sentence

The sentence has been corrected (line 245).

  • Line 237-241: According to line 50-51, MBC seems to be a subtype of TNBC. Then, is it appropriate to compare the patient survival between TNBC and MBC?

We think this is a relevant comparation, as we are stating that MBC is a more aggressive subtype of TNBC, with poorer prognosis.

Reviewer 3 Report

This is a comprehensive molecular study of this breast cancer subtype. I think it wold be improved though by a more detailed analysis of the distribution and type of mutations for some of the key oncogenes.

Author Response

This is a comprehensive molecular study of this breast cancer subtype. I think it wold be improved though by a more detailed analysis of the distribution and type of mutations for some of the key oncogenes.

We would like to thank the reviewer for the helpful suggestions and comments.

We have added the type of mutations for TP53 and PIK3CA on Supplementary Tables S2 and S3.